# Variability in *Neogloboquadrina pachyderma* stable isotope ratios from isothermal conditions: implications for individual foraminifera analysis

Lukas Jonkers[1], Geert-Jan A. Brummer[2], Julie Meilland[1], Jeroen Groeneveld[3] and Michal Kucera[1]

*Correspondence to*: Lukas Jonkers (ljonkers@marum.de)

[1] MARUM Center for Marine Environmental Sciences, University of Bremen, Bremen, Germany
[2] Royal Netherlands Institute for Sea Research, Texel, the Netherlands
[3] Department of Geology, Hamburg University, Hamburg, Germany

**Abstract.** Individual foraminifera analysis (IFA) holds promise to reconstruct seasonal to interannual oceanographic variability. Even though planktonic foraminifera are reliable recorders of environmental conditions on a population level, whether they also are on the level of individuals is unknown. Yet, one of the main assumptions underlying IFA is that each specimen records ocean conditions with negligible noise. Here we test this assumption using stable isotope data measured on groups of four shells of *Neogloboquadrina pachyderma* from a 16-19 days resolution sediment trap time series from the subpolar North Atlantic. We find a within-sample variability of 0.11 and 0.10 ‰ for $\delta^{18}O$ and $\delta^{13}C$ respectively that show no seasonal pattern and exceed water column variability in spring when conditions are homogeneous down to 100s of metres. We assess the possible effect of life cycle characteristics and delay due to settling on foraminifera $\delta^{18}O$ variability with simulations using temperature and $\delta^{18}O_{seawater}$ as input. These simulations indicate that the observed $\delta^{18}O$ variability can only partially be explained by environmental variability. Individual *N. pachyderma* are thus imperfect recorders of temperature and $\delta^{18}O_{seawater}$. Based on these simulations, we estimate the excess noise on $\delta^{18}O$ to be $0.11 \pm 0.06$ ‰. The origin and nature of the recording imprecision require further work, but our analyses highlight the need to take such excess noise into account when interpreting the geochemical variability among individual foraminifera.

**Short summary.** The variability in the geochemistry among individual foraminifera is used to reconstruct seasonal to interannual climate variability. This method requires that each foraminifera shell accurately records environmental conditions, which we test here using a sediment trap time series. Even in the absence of environmental variability, planktonic foraminifera display variability in their stable isotope ratios that needs to be considered in the interpretation of individual foraminifera data.

**1 Introduction**

Planktonic foraminifera hold the promise to provide palaeo-environmental information at high temporal resolution, owing to their life cycle, which is in the order of weeks to months and calcification that takes place over hours to days. This potential is exploited in individual foraminifera analysis (IFA), when instead of measuring groups of shells, shells are measured individually and the variability among the individual shells is used to reconstruct environmental variability during deposition of the sample. This approach has been applied to reconstruct changes in intra- and inter-annual ocean variability across time scales (Ganssen et al., 2011; Leduc et al., 2009; Rustic et al., 2015).

The use of IFA to reconstruct past oceanographic variability implicitly assumes that each foraminifera shell is a perfect recorder of environmental conditions during calcification and that there is no, or negligible, biological noise in this recording. The assumption of perfect recording seems reasonable because at population level temperature exerts a dominant control on foraminifera $\delta^{18}O$ and Mg/Ca (Bemis et al., 1998; Elderfield and Ganssen, 2000). Analytical issues aside (Fehrenbacher et al., 2020), the uncertainty associated with IFA is often viewed from the perspective of whether the population is well enough characterised, how habitat tracking may affect the results or how variability at different time scales (seasonality/ENSO) can be distinguished (Glaubke et al., 2021; Leduc et al., 2009; Metcalfe et al., 2020; Thirumalai et al., 2013) and only few consider calibration issues associated with individual planktonic foraminifera as a source of uncertainty (Glaubke et al., 2021).

However, there are several indications suggesting that whilst temperature exerts a first order control on the Mg/Ca and $\delta^{18}O$ of foraminifera, other factors (biotic and/or abiotic) also play a role. For instance, the variability in Mg/Ca and $\delta^{18}O$ in foraminifera populations from sediment samples often exceeds the variability that can be expected based on local hydrography (Groeneveld et al., 2019; Leduc et al., 2009). Whilst such evidence from sediment may be ambiguous due to uncertainty in the age of the sample and the exact habitat of the foraminifera analysed, laboratory studies also suggest that foraminifera geochemistry is affected by temperature-independent variability (Dueñas-Bohorquez et al., 2011; de Nooijer et al., 2014; Spero and Lea, 1993). Laboratory-based calibrations of $\delta^{18}O$-temperature relationships hint at a similar non-temperature related noise (Bemis et al., 1998; Erez and Luz, 1982). Observations from plankton nets and sediment traps also demonstrate marked variability (Davis et al., 2020b; Haarmann et al., 2011; Livsey et al., 2020). These observations are not conclusive in their own right, but together they suggest that there are reasonable grounds to assess if the composition of individual foraminifera can be used as a reliable environmental indicator.

Here we assess the variability in $\delta^{18}O$ and $\delta^{13}C$ among shells of *Neogloboquadrina pachyderma* collected in the subpolar North Atlantic Ocean using a moored sediment trap. The advantage of using sediment trap material is that the temporal origin of the shells is much better constrained than in sedimentary material (days to weeks compared to years to centuries) and that seasonal variability in the abundance of foraminifera does not affect the geochemical variability within each sample. Previous work on this time series has shown that on a population level *N. pachyderma* faithfully tracks the seasonal cycle in upper ocean temperature at this location (Jonkers et al., 2010). The site in the Irminger Sea serves as a natural laboratory because of deep wintertime mixing that makes the water column homogeneous down to 100s of metres. In this study we reanalyse the previously

published data with the specific aim to assess the variability in the stable isotope ratios and to what degree the observed variability can be explained by variability in the environment. We observe marked variability in $\delta^{18}O$ and $\delta^{13}C$ even at times when the water column was thoroughly mixed. We use a simple model to evaluate the influence of life cycle characteristics on foraminifera $\delta^{18}O$ variability and find that the observed variability exceeds predictions. Our simulations provide a first-order quantification of the excess $\delta^{18}O$ variability and we argue that this biological noise should be considered when interpreting the variability in $\delta^{18}O$ among individual foraminifera.

## 2 Material and methods

### 2.1 Sediment trap mooring setting

We analyse stable oxygen and carbon isotope data from *N. pachyderma* from a 2.5-year long sediment trap time series from the centre of the Irminger Gyre (ca. 59.25° N, 38.66° W; Fig. 1). The sediment trap was positioned at a water depth of 2750 m, 250 m above the bottom. Collecting intervals were 19 days from autumn 2003 to autumn 2004 and 16 days from autumn 2005 to summer 2007. During the year, temperature, which is the main control on $\delta^{18}O$ at this location (Jonkers et al., 2010), varies between approximately 5 and 10 °C near the surface (Fig. 1). There is no marked seasonal cycle in temperature from around 200 m depth, where temperatures remain at approximately 5 °C year-round. Deep convective mixing, resulting in isothermal conditions, takes place in winter time (de Jong et al., 2012). The time series of *N. pachyderma* stable isotopes we analyse here captures these isothermal conditions three times.

### 2.2 Data

Stable isotope measurements were performed on groups of four *N. pachyderma* shells (150-250 μm) with up to six measurements per collection interval. In Jonkers et al. (2010) we presented average stable isotope data, but here we return to the raw data and assess the variability within each sample. Even though the measurements were done on groups of four shells, the replicate measurements on small numbers of shells allow us to obtain a first order estimate of the minimum stable isotope variability within the population of *N. pachyderma*. Our analyses are therefore meaningful for the interpretation of IFA results. Not all samples from the time series contained enough shells of *N. pachyderma* (Fig. 1), so the complete data set consists of 172 measurements from 45 samples, of which 163 are from 36 samples with at least two measurements. All measurements were done using a Thermo MAT253 mass spectrometer coupled to a Kiel IV device. The analytical error (1 s.d.), determined from repeat measurements of the NBS-19 standard, amounts to 0.05 ‰ for $\delta^{18}O$ and 0.03 ‰ for $\delta^{13}C$. Further details about the mooring and the analytical procedure are presented in Jonkers et al. (2010).

The number of replicate measurements per sample is relatively low compared to what is used for IFA on sedimentary material. This is however justified given the short collection intervals of sediment trap samples (in our case 16-19 days) compared to the long integration time of sediment samples (at least decades to centuries). Moreover, with low numbers of measurements we are likely to underestimate the variability at population level and our inferences will therefore be conservative.

In order to obtain a conservative estimate of the variability among the measured groups of *N. pachyderma* shells we remove possible outliers. Given the small sample sizes, outliers were identified using all data in Fig. 2, and excluded from our analysis to avoid unnecessary inflation of inter-specimen variability. We calculated the residual from the mean for each sample and defined outliers as being more than 1.5 times the interquartile range away from the overall mean (Fig. 3). This approach resulted in the removal of 10 (6%) and 4 (2%) measurements of $\delta^{18}O$ and $\delta^{13}C$, respectively.

We compare the observations to expected $\delta^{18}O$ equilibrium values and estimates of the $\delta^{13}C$ of dissolved inorganic carbon ($\delta^{13}C_{DIC}$). We calculate equilibrium $\delta^{18}O$ ($\delta^{18}O_{eq}$) using the Kim and O'Neil (1997) palaeotemperature equation because *N. pachyderma* calcifies without an offset from this equation (Jonkers et al., 2010, 2013). For the deployments from 2003-2004 and 2005-2006 we use the same temperature and salinity data as in previous work (2010, 2013). However, for the deployment from 2006-2007 temperature and salinity data at 10 and 266 m are available from the nearby CIS mooring (59.66° N; 39.66° W) and we use these as it allows using *in-situ* surface salinity measurements and because of better temporal coverage at depth (Jonkers et al., 2016). Seawater $\delta^{18}O$ ($\delta^{18}O_{seawater}$) was derived from salinity, using the regional salinity-$\delta^{18}O_{seawater}$ relationship used in Jonkers et al. (2010).

Estimates of $\delta^{13}C_{DIC}$ are the same as in Jonkers et al. (2013) and based on multiple-linear regression of temperature, salinity and nutrients within the wider subpolar North Atlantic. Since the $\delta^{13}C_{DIC}$ data are derived from data that represent long-term average conditions (climatology), they cannot be used to the same level of detail as $\delta^{18}O$. We compare the measured variability in $\delta^{13}C$ to the seasonal range in $\delta^{13}C_{DIC}$ and the seasonal range in expected foraminifera $\delta^{13}C$ by taking into account a temperature-dependent offset from $\delta^{13}C_{DIC}$ (Jonkers et al., 2013).

## 2.3 Predicting *N. pachyderma* $\delta^{18}O$ variability

Planktonic foraminifera intermittently add chambers during their life cycle and start sinking towards the ocean floor upon death. The signal contained in their stable isotope ratios is therefore a reflection of the environmental conditions during a certain time prior to arrival in the sediment trap. To assess if the observed variability in $\delta^{18}O$ can be explained by temperature and $\delta^{18}O_{seawater}$ alone, we predict $\delta^{18}O$ calcite ($\delta^{18}O_{equilibrium}$) using a model that is more complex in its representation of calcification than what is usually attempted when interpreting results of individual foraminifera analyses (Glaubke et al., 2021; Groeneveld et al., 2019; Thirumalai et al., 2013). We simulate foraminifera $\delta^{18}O$ as an average of chamber $\delta^{18}O$ and add a delay between formation of the final chamber and arrival at the sediment trap that reflects time spent in the water column without calcification and sinking to the depth of the trap. In this way we represent calcification during the foraminifera life cycle more realistically and allow for more variability than when assuming that each foraminifera shell represents environmental conditions averaged over one (calendar) month. Our approach is based on the following assumptions: 1) foraminifera build their chambers at random times during their life cycle; 2) chamber formation takes one day; 3) each foraminifera shell consists of four chambers with equal mass and 4) all shells have the same mass.

The first assumption is reasonable in light of the limited amount of information available on the (temporal aspects of the) ontogeny of *N. pachyderma* (Bé et al., 1979; Spindler, 1996). The assumed duration of chamber formation is based on culture studies (Bé et al., 1979; Spindler, 1996). However, culture studies in the closely related species N. dutertrei have shown that chamber formation may take up to four days (Fehrenbacher et al., 2017). Longer chamber formation could in theory reduce the variability foraminifera $\delta^{18}O$ because of increased smoothing of the environmental signal. In practice this effect is however negligible because of strong temporal autocorrelation in the $\delta^{18}O_{equilibrium}$ time series that renders the effect of smoothing of up to four days insignificant. Our approach thus yields an estimate of variability that is robust against the likely range of chamber formation duration. In *N. pachyderma* the last whorl of the shell makes up most of the mass and generally consists of four chambers that are of similar size. The assumed number and equal mass of the chambers is thus reasonable. The last assumption is out of convenience.

For each sample we simulate $\delta^{18}O$ for different calcification spans (the time it takes to form a four-chambered synthetic shell) and delays (the time between formation of the last chamber and arrival at the trap). We vary the calcification span between 4 and 168 days and the delay between 5 and 180 days. The minimum value for the delay is based on estimates of sinking velocity of planktonic foraminifera (Takahashi and Bé, 1984). We exclude scenarios where the sum of calcification span and delay is more than 181 days because of the clear seasonal pattern in mean $\delta^{18}O$. This pattern indicates that long delays are unlikely because minimum $\delta^{18}O$ values are observed shortly after peak temperatures. Very long calcification spans are also unlikely as these would result in small seasonal $\delta^{18}O$ variation. We allow for some variability in the calcification span and delay by varying the calcification span in each scenario within a lognormal distribution with the mode equal to the calcification span and a standard deviation of 0.3. The delay is varied using a normal distribution with a standard deviation that is the square root of the delay.

To investigate the effect of calcification depth we run two groups of simulations, one where we assume that calcification takes place exclusively at the surface and another where we allow for variable calcification depth, either near the surface or at depth (ca. 250 m), within each sample. We include the possibility that shells were formed at depth because *N. pachyderma* is known to inhabit a wide depth range (Greco et al., 2019) and previous studies indicated a large and variable apparent calcification depth (Kohfeld et al., 1996; Simstich et al., 2003). However, the real range of apparent calcification depth of *N. pachyderma* in the Irminger Sea is probably narrower than the 200-250 m assumed in the simulations. This is because the average $\delta^{18}O$ of *N. pachyderma* shows a seasonal trend with a magnitude that suggests an apparent calcification depth around 50 m (Jonkers et al., 2010, 2013). This scenario thus likely overestimates variability, especially during the summer season when the water column is stratified. We do not simulate calcification exclusively at depth because this is clearly at odds with observed seasonal amplitudes of $\delta^{18}O$ and $\delta^{13}C$.

We do not consider the possibility of ontogenetic vertical migration in our simulations. This is partly an assumption out of necessity because we do not have temperature and salinity data between the surface and 200 m depth for the entire time series. We however stress that our approach is conservative because ontogenetic migration would decrease the variability in foraminifera stable isotope ratios.

To be consistent with the measurements on groups of four shells,  we average the $\delta^{18}O$ of four simulated shells. We add measurement uncertainty (white noise with a standard deviation of 0.05 ‰) to the averaged $\delta^{18}O$ and calculate the standard deviation of the $\delta^{18}O$ of 2-6 groups (depending on the sample) of four shells. We repeated this process 300 times for each sample and for each combination of delay and calcification span. We consider cases significant when the predicted standard deviation is higher than the observed standard deviation in 95 % of the simulations.

Estimates of $\delta^{18}O_{equilibrium}$ are not available for the entire time series and our simulations are therefore restricted to the spring of 2004, the spring to autumn of 2006 and the spring of 2007. Because we lack detailed data on $\delta^{13}C_{DIC}$ we did not simulate foraminifera $\delta^{13}C$. We, however, do not ignore foraminifera $\delta^{13}C$ in our analysis.

Modelling is by definition a simplification of reality. Even though important aspects of our model (variable depth, faster calcification) yield estimates of expected variability that are higher than in previous work, we follow previous work and consider local temperature and $\delta^{18}O_{seawater}$ as the only predictors of $\delta^{18}O_{equilibrium}$ (Glaubke et al., 2021; Thirumalai et al., 2013). For simplicity we do not consider advection of foraminifera because it is not directly clear how advection within the Irminger Gyre, where temperatures are spatially rather uniform, would influence the temperature variability that planktonic foraminifera would be exposed to during calcification. Assessing the influence of advection can only be done using lagrangian modelling (van Sebille et al., 2015) and ultimately relies on the accuracy with which the model captures spatial and temporal temperature variability. Such modelling is beyond the scope of this study. We neither consider the effect of the carbonate ion concentration ($[CO_3^{2-}]$) on foraminifera stable isotopes (Spero et al., 1997). Because of the positive correlation between temperature and $[CO_3^{2-}]$(Jonkers et al., 2013) and a negative correlation between $[CO_3^{2-}]$and foraminifera $\delta^{18}O$ (Spero et al., 1997) the $[CO_3^{2-}]$ effect would slightly increase the seasonal range $\delta^{18}O_{equilibirum}$. Assuming that the sensitivity of *N. pachyderma* $\delta^{18}O$ is similar to that of *G. bulloides*, the increase would be in the order of 0.15 ‰. Since we do not consider this possible source of variability, our simulations are likely to provide conservative estimates of foraminifera $\delta^{18}O$ variability.

## 3 Results and discussion
### 3.1 Raw data
The $\delta^{18}O$ of *N. pachyderma* varies between 0.93 ‰ in early winter 2006 and 2.88 ‰ in spring 2004 (Fig. 2). The overall seasonal amplitude is around 1 ‰, with a minimum in $\delta^{18}O$ that lags the maximum temperatures by one to two months. Stable oxygen isotope ratios are in general within the range of predicted $\delta^{18}O_{equilibrium}$. The $\delta^{13}C$ values show a smaller amplitude (-0.37 to 0.58 ‰) and are always offset from $\delta^{13}C_{DIC}$ (Fig. 2). The $\delta^{13}C$ values generally decrease from spring to winter. For both $\delta^{18}O$ and $\delta^{13}C$ the observed within sample variability exceeds the analytical uncertainty (Fig. 3).

After outlier removal, the within-sample range of $\delta^{18}O$ varies between 0.05 and 0.51 ‰ (mean 0.24 ‰) and does not show a consistent pattern during the year (Fig. 4). There is no relationship between the number of measurements within a sample and the range in $\delta^{18}O$ (Fig. 4). The within sample range is always smaller than

the seasonal range in surface $\delta^{18}O_{equilibrium}$. Most of the time the observed $\delta^{18}O$ range is also smaller than the vertical gradient in $\delta^{18}O_{equilibrium}$, except during isothermal conditions in spring when it exceeds the $\delta^{18}O_{equilibrium}$ range (Fig. 4). The range in $\delta^{13}C$ is similar to $\delta^{18}O$ and varies between 0.06 and 0.46 ‰ (mean 0.21 ‰) and neither shows a clear seasonal pattern (Fig. 4). Compared to $\delta^{18}O$, the range of foraminifera $\delta^{13}C$ is more often above the expected range (Fig. 4).

There are two important points regarding these initial observations. The first is that the observed range in foraminifera stable isotope values exceeds the expected range in spring (April - May) when the water column is well-mixed down to 800 m depth. This variability arises from apparently random positive and negative offsets from $\delta^{18}O_{equilibrium}$, suggesting that it does not result from a mechanism that would cause a systematic bias in the foraminifera $\delta^{18}O$. Advection or long foraminifera life spans, which could theoretically cause foraminifera from the previous summer to survive until spring, are therefore unlikely to provide a full explanation for the observed variability. This is the first indication that the variability in foraminifera isotope ratios does not solely result from environmental variability. The second observation is the apparent lack of a seasonal cycle in the range in $\delta^{18}O$ and $\delta^{13}C$ even though stratification develops as the sea surface warms. In theory, the variability in foraminifera stable isotope ratios could therefore increase towards the warm season. The fact that this cannot be seen in the data indicates that *N. pachyderma* calcifies in a relatively narrow and constant vertical range.

**3.2 Predicted foraminifera $\delta^{18}O$ variability**

To assess if observed variability in $\delta^{18}O$ of *N. pachyderma* is higher than the variability expected from temperature and $\delta^{18}O$ of seawater at the time of sampling because the foraminifera calcified prior to the sampling we carried out simulations using a range of possible calcification spans and delays. These simulations indicate that the standard deviation of *N. pachyderma* $\delta^{18}O$ in spring when the water column is virtually isothermal (IRM-1 A-14, IRM-3 A-13, IRM-3 A-14, IRM-4 A-14 and IRM-4 A-15) exceeds what can be expected based on reasonable calcification histories and delays (Fig. 5). The predicted variability only significantly exceeds the observations during summer and almost exclusively in the simulations that allow variable calcification depth. Our simulations are thus sensitive to the choice of calcification depth and it is important to assess if the scenario with variable depth habitat is more realistic than the scenario with constant, near-surface habitat. We can compare both scenarios by determining the prediction error in the mean $\delta^{18}O$ across all samples (Fig. 6). The minimum prediction error is, in both scenarios, distributed along an arc shape, with lower errors at longer calcification spans and delays up to about a month or at short calcification spans and delays in the order of one to two months. However, the errors reach markedly lower values in the scenario where calcification only occurs near the surface. Because the seasonal peak in temperature is reached earlier at the surface than at depth, it remains difficult to determine precisely which combination of calcification depth, calcification span and delay is most realistic, but the amplitude of the mean seasonal $\delta^{18}O$ indicates that the surface only scenario is closer to what the foraminifera actually experienced than the variable depth scenario. This indicates that even when taking reasonable calcification histories and delays into account, the observed variability in foraminifera $\delta^{18}O$ is unlikely to reflect environmental (temperature) variability alone.

Our simulations also permit us to put some constraints on the calcification span and delay that characterises *N. pachyderma* at this location. The hardest constraints can be put on the possibility of long delays between formation of the last chamber and arrival at the trap. Sinking speed measurements suggest that the delay due to sinking at this location is likely to be between 5 and 19 days (Takahashi and Bé, 1984). We obtain minimum prediction errors for delays up to approximately two months (Fig. 6). Subtracting the sinking time estimates from these delays implies that *N. pachyderma* is unlikely to spend more than one month in the water column without calcifying after the last chamber has formed. This means that the simulations with delays >100 days are not realistic.

Our simulations indicate that calcification spans under two weeks yield smaller errors when associated with delays in the order of 30 - 60 days and similarly low prediction errors are obtained using longer calcification spans and shorter delays. Based on our data it is difficult to ascertain which cases are more realistic. However, such long delays would require long intervals spent in the water column without calcification. A single culture study using Antarctic *N. pachyderma* showed intermittent chamber formation over a period of about two months and a single case of gametogenesis approximately two weeks after the formation of the final chamber (Spindler, 1996). Other studies also suggest an approximately two month life span (Davis et al., 2017, 2020a). This suggests that delays of up to approximately one month (including settling) and calcification of the final four chambers over the course of about two months are most probable.

### 3.3 Excess foraminifera $\delta^{18}O$ variability

The mean observed standard deviation for of $\delta^{18}O$ is $0.11 \pm 0.05$ ‰ for the complete time series and $0.10 \pm 0.03$ ‰ for the samples from the time when the water column was isothermal (IRM-1 A-14, IRM-3 A-13, IRM-3 A-14, IRM-4 A-14 and IRM-4 A-15). As noted above, the fact that the variability in $\delta^{18}O$ does not show a consistent pattern during the year, suggests that we have captured the full range of within-sample variability even though the number of measurements per sample is relatively low. Since our measurements are based on groups of four shells the observed standard deviation is an underestimate of the standard deviation among individual shells. Assuming that each shell in the group contributed equally to the total mass, the degree of underestimation of the standard deviation scales with the square root of the group size (Groeneveld et al., 2019). Thus we multiply the observed standard deviation by two ($\sqrt{4}$) to obtain an estimate of the standard deviation of individual shells. That means that the $\delta^{18}O$ of individual foraminifera at this location is likely to have a standard deviation of $0.19 \pm 0.07$ ‰ ($0.21 \pm 0.11$ ‰ when considering all observations).

For the samples from the times when the water column was deeply mixed, i.e. when variations in temperature, salinity and hence $\delta^{18}O_{equilirbium}$ were negligible, our simulations predict a standard deviation for individual shells of 0.08 ‰. This prediction is identical for both depth scenarios. It includes a 0.05 ‰ measurement uncertainty and is based on all considered scenarios with a delay less than 100 days, which is reasonable given the low model skill at longer delays. Assuming that our simulations are a reasonable approximation of reality, the excess variability (s.d.) that cannot be explained by variability in temperature and $\delta^{18}O_{seawater}$ is therefore $0.11 \pm 0.06$ ‰, which in terms of temperature roughly translates to a standard deviation of 0.4 °C.

Whereas our modelling approach provides an estimate that is likely closer to reality than assuming that foraminifera reflect environmental conditions averaged over a single (calendar) month, our estimate could be evaluated by simulating other calcification trajectories. We found that our results are insensitive to the duration of chamber formation and experiments where we allowed complete shell formation within one day, equivalent to assigning all weight to the last chamber, yielded an expected standard deviation of individual foraminifera $\delta^{18}O$ of 0.09 ‰. Therefore, the assumption of equal weight of the four chambers has little bearing on our results. Ultimately, the modelled foraminifera $\delta^{18}O$ depends on the hydrographic data used to estimate $\delta^{18}O_{equilibrium}$. By using data from the surface and from great depth, we have obtained two end-member scenarios of vertical $\delta^{18}O_{equilibrium}$ variability that implicitly encompass ontogenetic vertical migration. However, future estimates of expected individual foraminifera $\delta^{18}O$ variability could be improved by explicitly incorporating horizontal $\delta^{18}O_{equilibrium}$ variability and advection during shell growth in the modelling strategy.

Apart from being sensitive to our modelling design and data availability, our estimate of excess $\delta^{18}O$ variability among individual shells is also sensitive to the quantification of variability among shells. To obtain a conservative estimate we excluded potential outliers. Were we to consider all measurements, the average standard deviation among groups would be $0.15 \pm 0.11$ ‰ ($0.17 \pm 0.09$ ‰ during spring) and the resulting excess $\delta^{18}O$ variability $0.25 \pm 0.19$ ‰. Thus our approach yields a conservative and better constrained estimate of the excess variability.

We compare this estimate of unexplained $\delta^{18}O$ variability to two studies that used individual foraminifera $\delta^{18}O$ from cores in the eastern equatorial Pacific Ocean to infer changes in the El-Niño Southern Oscillation. In the first study, the range in the standard deviations of *N. dutertrei* $\delta^{18}O$ shells in eight time slices across the past 50,000 years amounts to 0.15 ‰ (Leduc et al., 2009). In the second study, Rustic et al. (2015) interpreted changes in the standard deviation of *G. ruber* $\delta^{18}O$ over the last millennium that were smaller than 0.45 ‰ (variance of 0.20 ‰$^2$). Forward modelling studies also indicate that changes in the amplitude (doubling or halving) in the central equatorial Pacific would translate to changes in the standard deviation of IFA of maximum 0.15 ‰ (Thirumalai et al., 2013). In all cases, the unexplainable $\delta^{18}O$ variability we observe makes up a substantial part of the signal. Thus, non-temperature effects on individual foraminifera $\delta^{18}O$ need to be considered when interpreting the results of IFA.

### 3.4 Possible causes of excess variability

The relatively constant variability in $\delta^{18}O$ and $\delta^{13}C$ within the *N. pachyderma* population in the Irminger Sea during the year argues against a direct environmental influence on the variability. This is because on seasonal time scales environmental variability is strongly correlated to temperature and/or stratification. The observed variability could therefore be random or reflect biological processes within the population of foraminifera, where each shell, or each chamber, records the environment with a small offset. As long as the excess variability remains random or uncorrelated with the environment, the average stable isotope composition of (large enough subsample of) a foraminifera population will accurately reflect environmental conditions. On a population level, planktonic foraminifera $\delta^{18}O$ is indeed a reliable indicator of seawater temperature and $\delta^{18}O_{seawater}$ (e.g. Bemis et

al., 1998; Erez and Luz, 1982), suggesting that the excess variability among individual specimens is cancelled out within populations.

Alternatively  the excess variability could arise from environmental or biotic forcing that we did not consider in our simulations. Crucially, any possible mechanism needs to explain the approximately equal variability in $\delta^{18}O$

and $\delta^{13}C$ that we observe in the time series.

Shell size is likely to affect metabolic rates and the observed excess variability could therefore be related to differences in shell size (Spero and Lea, 1993, 1996). However, in such a scenario, the effect would be expected to be much stronger on $\delta^{13}C$ than on $\delta^{18}O$, as is the case for *G. bulloides* (Spero and Lea, 1996). The comparable

variability in both carbon and oxygen isotope ratios thus suggests that size differences within the foraminifera population are unlikely to explain the observed excess variability.

Along similar lines, growth rate may vary among individual foraminifera and thereby influence the stable isotope composition, as has for instance been shown for corals (McConnaughey, 1989). However, in corals,

$\delta^{13}C$ is, like with the size effect above, more sensitive to changes in the growth rate than $\delta^{18}O$. Therefore, if such an effect were to occur among (non-symbiotic) planktonic foraminifera, growth rate differences  would neither be the likely cause of the excess variability in stable isotope ratios.

The excess variability could also arise from differences in the proportion of crust to lamellar calcite. We did not

perform a systematic analysis of the degree of encrustation of *N. pachyderma* in the sediment trap samples,but in the many years of work on this time series we have never come across a crust-free specimen.  It is nevertheless likely that the degree of encrustation varies among individuals and variable crust to lamellar calcite ratios among foraminifera could therefore add temperature-independent noise, similar to what has been suggested for Mg/Ca (Jonkers et al., 2016, 2021). However, the difference between crust and lamellar calcite

$\delta^{18}O$ of *N. pachyderma* intercepted in spring when the water column was well-mixed is not significant (Livsey et al., 2020). Variable encrustation can therefore not be the explanation for the excess $\delta^{18}O$ variability observed during the isothermal conditions in spring. In addition, this explanation would require that the crust and lamellar calcite also have different carbon isotope ratios. However, previous work is inconclusive in this regard. Observations from plankton hauls suggest that encrusted and crust-free *N. pachyderma* have systematically

different $\delta^{13}C$, but that the effect of encrustation is not as strong as on $\delta^{18}O$ (Kohfeld et al., 1996). A larger dataset from the sediment on the other hand, indicates no effect of encrustation (Healy-Williams, 1992). Whether or not variable encrustation is the cause of the observed excess variability in $\delta^{18}O$ and $\delta^{13}C$ therefore remains an open question.

Notwithstanding the fact that the exact cause of the excess variability in *N. pachyderma* stable isotope ratios needs to be constrained in future studies, our analysis shows that individual planktonic foraminifera record environmental conditions with less precision than average populations. Our study thus confirms earlier indications (Groeneveld et al., 2019; Livsey et al., 2020), but we have attempted a first quantification of this noise for $\delta^{18}O$, which has up to now been ignored in the interpretation of individual foraminifera data.

**3.5 Implications for reconstructions of environmental variability based on individual foraminifera**

The possibility that individual planktonic foraminifera record seawater conditions with limited precision has up to now been overlooked when using the geochemistry of individual planktonic foraminifera to reconstruct climate variability. Our analyses provide evidence that the $\delta^{18}O$ of individual *N. pachyderma* shells may reflect seawater temperature and $\delta^{18}O$ with a precision of only 0.11 ‰. For now we assume that the cause of this lack of precision is random biological noise, but future studies are needed to verify that this is indeed the case, or if the recording precision is dependent on environmental or biological factors.

Our observations strengthen the case to use large numbers of foraminifera, not just for IFA. Depending on instrumental precision the biological recording noise doubles or triples the variability that can be expected in (individual) planktonic foraminifera $\delta^{18}O$, even when temperatures were constant during calcification. Any study using individual foraminifera $\delta^{18}O$ to infer past environmental variability, thus needs to cross this noise threshold in order to obtain meaningful results. Lack of recording precision will also influence the shape of the distribution of IFA results (Fig. 7), especially at the tails of the distribution that are often used to infer changes in upper ocean dynamics (Glaubke et al., 2021).

There are no reasons to believe that the existence of biological recording noise is unique to *N. pachyderma* or to stable oxygen and carbon isotopes alone. In fact, most of the indications for excess variability are based on other species (Bemis et al., 1998; Erez and Luz, 1982; Leduc et al., 2009; Spero and Lea, 1996). We therefore presume that a similar noise characterises other species and proxies as well. However, more research is needed to constrain the nature and causes of this lack of precision in the recording by individual foraminifera. Future research, including culturing, needs to consider different species in different environmental settings. Including Mg/Ca as an independent temperature-sensitive parameter may also help to elucidate the cause of the excess variability. Notwithstanding, our data clearly show that the assumption that individual planktonic foraminifera are perfect recorders of (monthly mean) temperature is not valid. Biology cannot be ignored in the interpretation of planktonic foraminifera proxies.

**4 Conclusions**

Stable isotope measurements on groups of four shells of *N. pachyderma* from a 16-19 day resolution sediment trap time series in the subpolar North Atlantic show large within sample variability. Stable oxygen and carbon isotope ratios within the time series have a mean standard deviation of 0.11 and 0.10 ‰, respectively and show no relationship with the seasonal trend in temperature ($\delta^{18}O_{eq}$) or the $\delta^{13}C$ of dissolved inorganic carbon. This lack of a seasonal pattern in the variability suggests that at this location *N. pachyderma* has a seasonally rather stable apparent calcification depth, which based on the amplitude of the sample mean $\delta^{18}O$ is around 50 m. Due to deep mixing the site is characterised by homogeneous water column conditions at the start of the spring foraminifera flux pulse. *Neogloboquadrina pachyderma* stable isotope variability at this time exceeds the variability that can be expected from the local hydrography, indicating that an additional source of variability that has so far not been considered in the interpretation of records of the geochemistry of individual foraminifera. Predictions of the observed variability in *N. pachyderma* $\delta^{18}O$ from temperature and $\delta^{18}O_{seawater}$

using realistic calcification and settling histories fail to match the observed variability. We therefore conclude that the $\delta^{18}O$ of individual *N. pachyderma* imperfectly record temperature and $\delta^{18}O_{seawater}$. Whether random, or controlled by environmental or biological factors, *N. pachyderma* records environmental variability with some degree of noise.

Our first-order estimate of the recording noise of individual specimens amounts to 0.11 ‰ (1 sd), which is

approximately double the typical analytical noise. Whilst more studies are needed to constrain the origin and variability in this recording noise, there are no reasons to believe it is a feature exclusive to *N. pachyderma*. The considerable recording noise should therefore be considered when interpreting geochemical variability among individual foraminifera.

**Acknowledgements**

L.J. acknowledges funding through the German climate modelling initiative PALMOD, funded by the German Ministry of Science and Education (BMBF). J.M. is funded by the Cluster of Excellence „The Ocean Floor – Earth's Uncharted Interface" funded by the German Research Foundation (DFG). The CIS mooring temperature and salinity data were collected and made freely available by the OceanSITES project (www.oceansites.org) and

the national programs that contribute to it. We acknowledge funding within the VAMOC (RAPID) program NWO grant 854.00.020 for deployment of the sediment trap moorings.

**Data availability statement**

The stable isotope data have been submitted to pangaea.de


**Author contributions**

LJ concept, analysis. Modelling concept with feedback from JM

GJB mooring, funding

LJ led the writing of the manuscript and created the figures. All authors reviewed and edited the manuscript.


**Competing interests**

The authors declare that they have no conflict of interest.


**Figures**

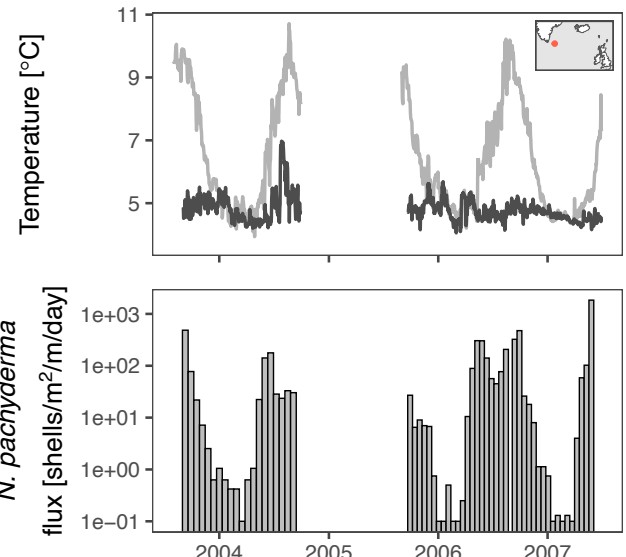

**Fig.1**: Temperature at the surface and at 200-250 m water depth at the Irminger Sea sediment trap mooring (red

dot in map inset). In winter and spring the water column is mixed to great depths

Bottom panel shows the evolution of the shell flux of *N. pachyderma* (150-250 μm from Jonkers et al. (2010); zero fluxes are shown as 0.1 shells/m²/day); stable isotope data are available for all but lowest flux intervals (Fig. 2). No data is available for the deployment from 2004 to 2005 because of failure of the sediment trap.

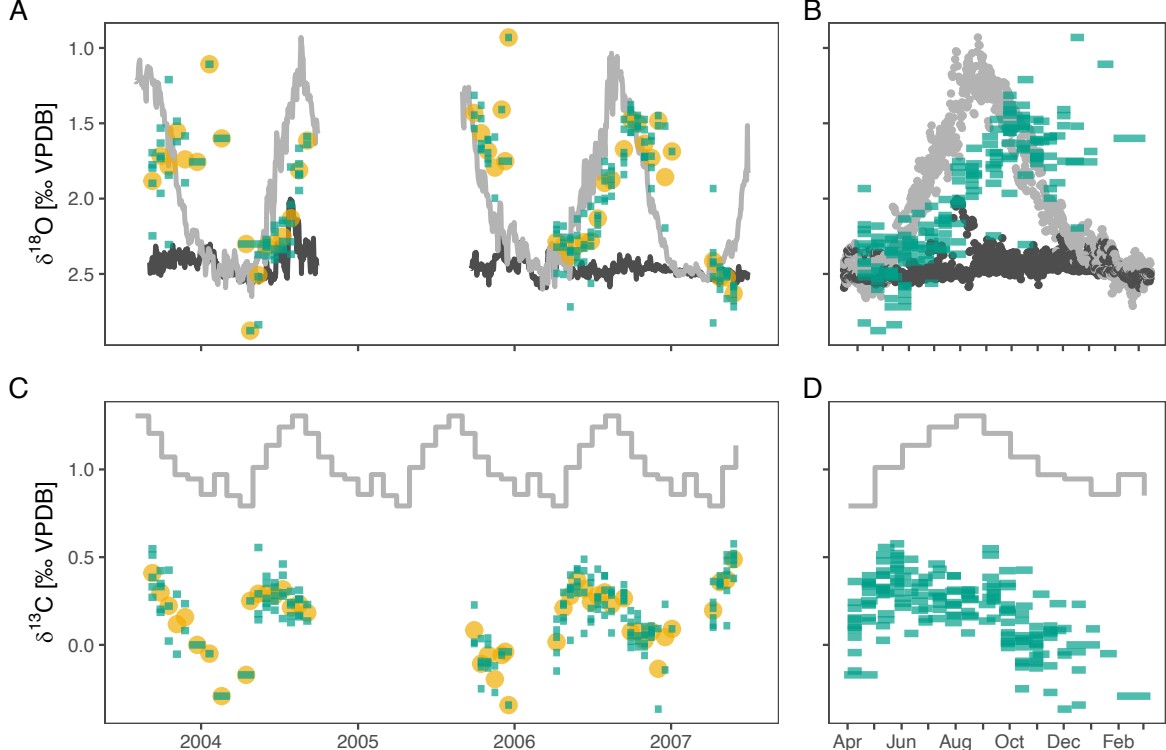

**Fig. 2**: *Neogloboquadrina pachyderma* stable isotopes in the Irminger Sea sediment trap time series. Panels A and C show the time series of $\delta^{18}O$ and $\delta^{13}C$, respectively. Panels B and D highlight the annual pattern, they show the same data collapsed onto a single year. Green bars extend over the collection interval and show individual measurements for groups of four shells; yellow points are average values per sample. The light grey lines depict surface $\delta^{18}O_{eq}$ and $\delta^{13}C_{DIC}$; dark grey lines in A and B are $\delta^{18}O_{eq}$ at 200-250 m depth. The oxygen and carbon isotopes show considerable variability within each sample, also when the water water column is mixed in April - May, suggesting stable isotope variability in excess of what can be explained based on environmental variability alone. The average oxygen isotope ratios track the seasonal cycle of near surface $\delta^{18}O_{eq}$ (light grey line in A and B) with an offset due to a slightly deeper calcification depth and/or a delay. Stable carbon isotopes also show a clear seasonal cycle, but with a marked offset from the $\delta^{13}C$ of DIC (grey line in C and D).

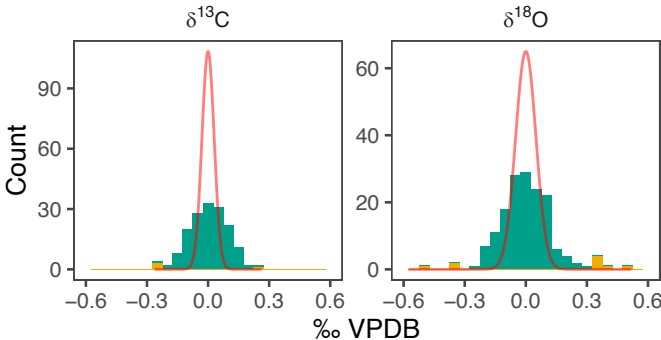

**Fig. 3**: Within-sample variability in *N. pachyderma* stable isotopes exceeds analytical noise. Histograms of residual δ18O and δ13C compared to expected density distribution if variability were due to analytical uncertainty alone (red line). Yellow colours indicate outliers (see methods).

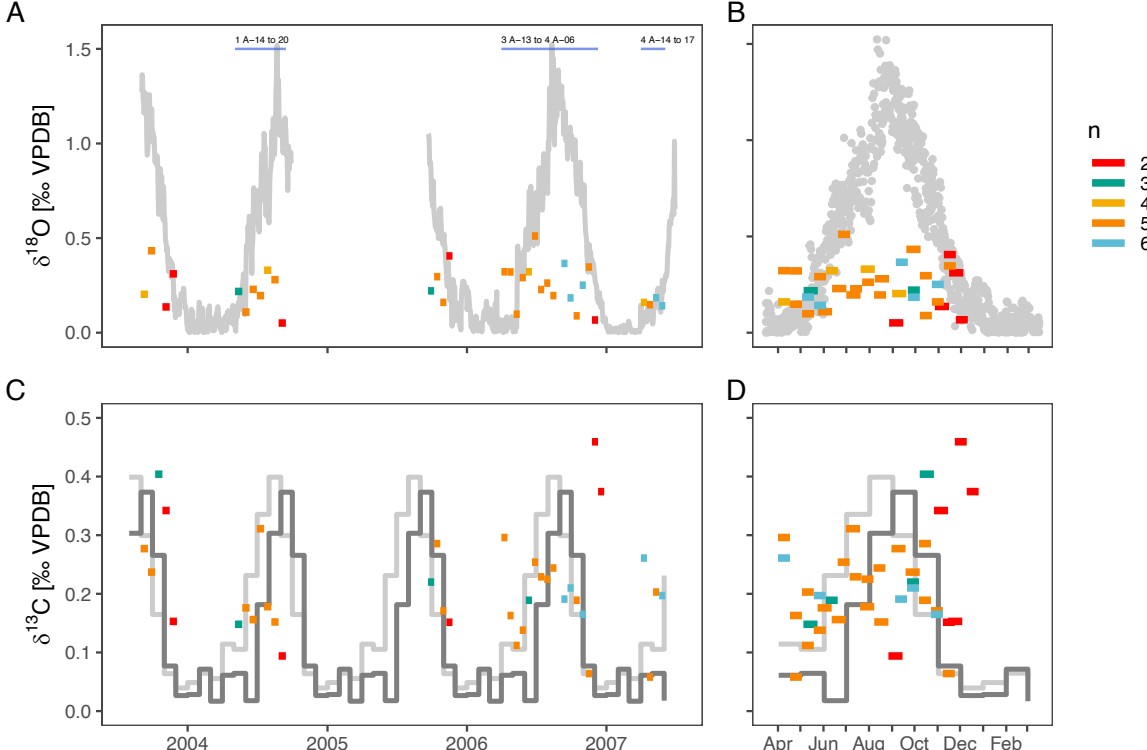

**Fig. 4**: The within-sample stable isotope range of *N. pachyderma* exceeds expected variability in spring when water column conditions are homogeneous and shows no consistent seasonal pattern. Note difference scales for $\delta^{18}O$ and $\delta^{13}C$. Bars extend to the collection intervals, colours indicate number of measurements per sample. Grey colours in A and B depict the difference in $\delta^{18}O$ between the surface and 200-250 m water depth. Light grey lines in C and D show the seasonal range in $\delta^{13}C_{DIC}$ and dark grey lines the seasonal range in foraminifera $\delta^{13}C$ calculated using a temperature-dependent offset from $\delta^{13}C_{DIC}$ (see methods). Samples for which the $\delta^{18}O$ variability is simulated (Fig. 5) are indicated in A.

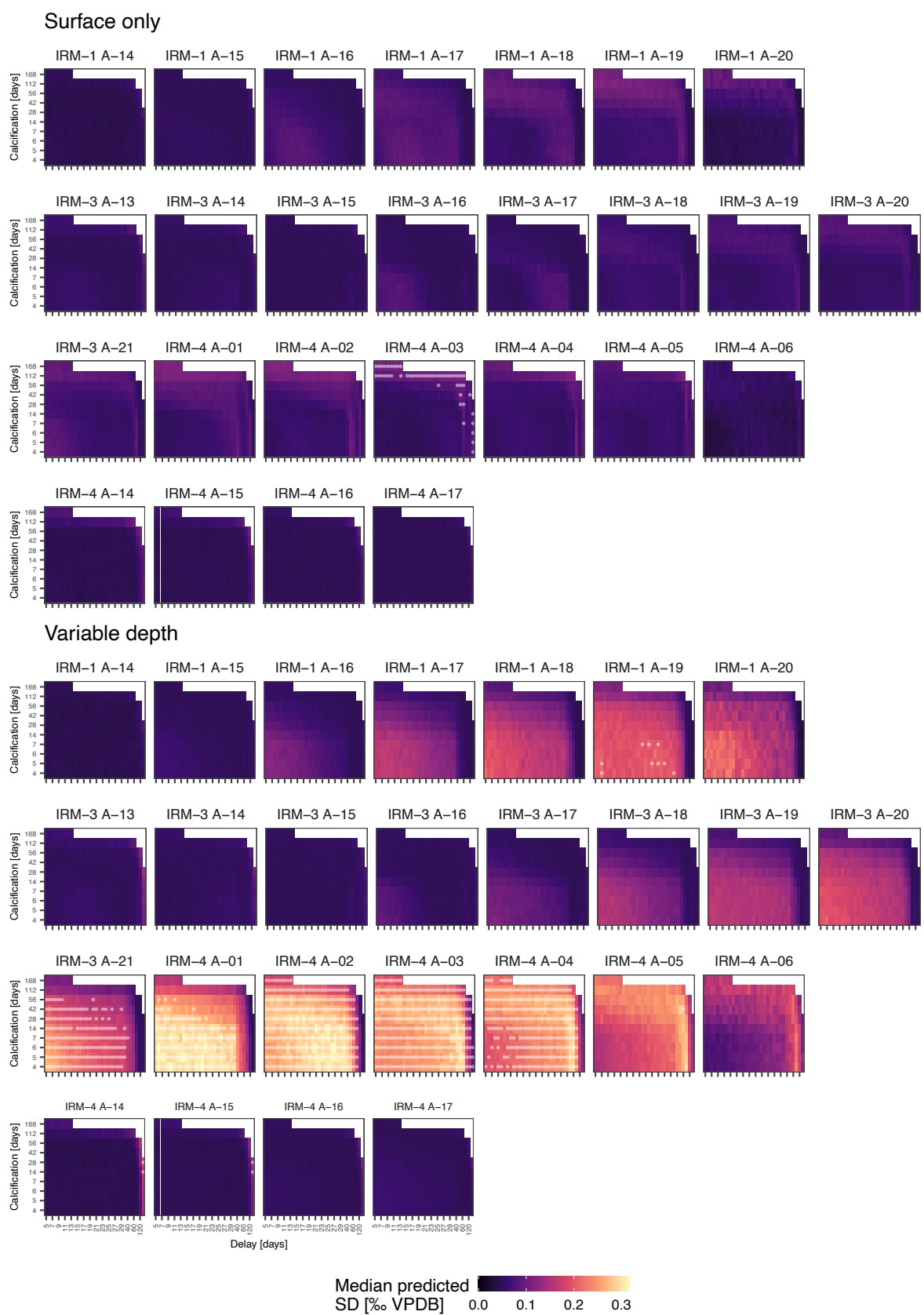

500

**Fig. 5**: Observed $\delta^{18}O$ variability in *N. pachyderma* generally exceeds expectations. Simulated $\delta^{18}O$ variability as a function of calcification span and delay for the surface only and variable depth scenarios for each sample indicated in Fig. 4. White dots indicate scenarios where the simulated variability significantly exceeds the

observed variability, note that this only occurs when a variable calcification depth is assumed. Samples are
505    ordered by year (with two rows for the 2005 - 2006 period), such that springtime samples are shown on the left.
Note that for clarity x axis ticks and labels are only shown for every second tick, all steps are shown in Fig. 6.

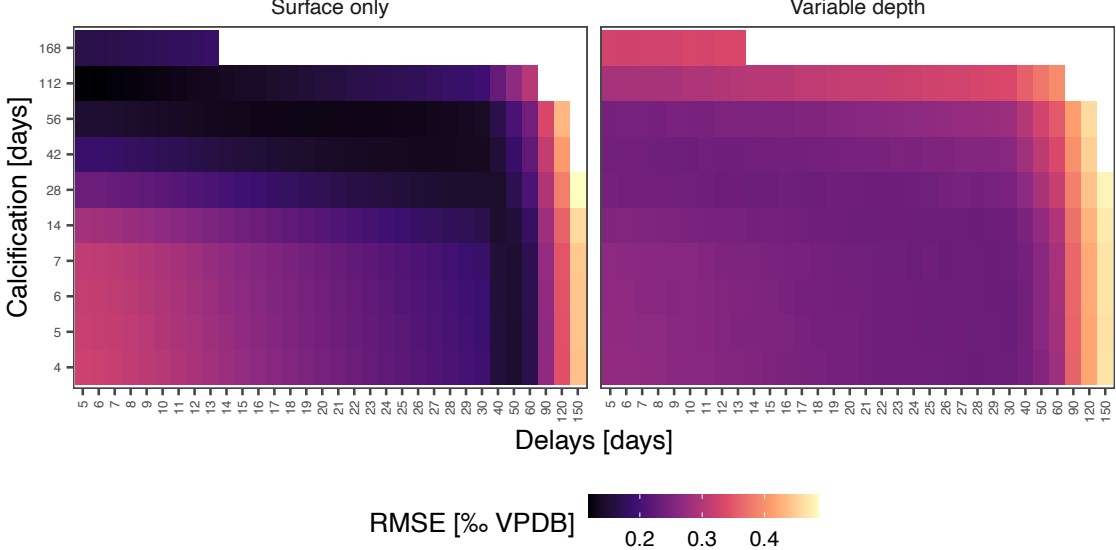

**Fig. 6**: Mean foraminifera $\delta^{18}O$ constrains simulations. Prediction errors for sample mean $\delta^{18}O$ reach markedly lower values for the surface-only simulations, indicating that this scenario is more likely to characterise *N. pachyderma* in the Irminger Sea. This means that the observed variability (Fig. 4) is unlikely a reflection of temperature and $\delta^{18}O_{seawater}$ variability alone and that the $\delta^{18}O$ of individual *N. pachyderma* shells is not a precise indicator of environmental conditions during calcification.

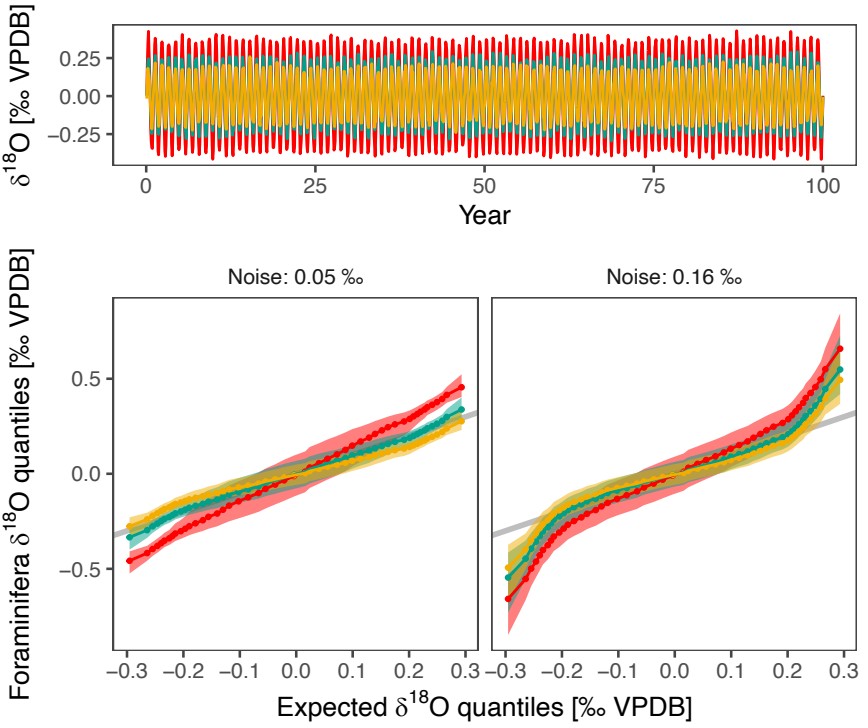

**Fig. 7**: Excess $\delta^{18}O$ variability mostly affects tails of $\delta^{18}O$ distribution within individual foraminifera. This simple simulation shows the effect of excess variability on capability to reconstruct changes in the amplitude of the seasonal cycle. The input consists of a synthetic $\delta^{18}O_{eq}$ time series with a seasonal amplitude of 0.25 ‰ that is not atypical of conditions in the central equatorial Pacific. The monthly time series is constructed using a sine wave with 0.02 ‰ random noise and is sampled 100 times at random to crudely represent planktonic foraminifera $\delta^{18}O$. This is an optimistic scenario as fewer foraminifera are usually used for IFA. The Q-Q plots show the effect of a change in the seasonal amplitude of $\delta^{18}O_{eq}$ for a scenario that only accounts for analytical noise (assumed to be 0.05 ‰) and for another that incorporates the excess variability found in this study. Higher noise levels affect the tails of the distribution and make it harder to detect changes in the seasonality.

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
