# Peer review of "Variability in *Neogloboquadrina pachyderma* stable isotope ratios from isothermal conditions: implications for individual foraminifera analysis"

_Climate of the Past, 2021_

## Author Comment (AC1)

We would like to thank the reviewer for their careful reading of our manuscript and their constructive comments. Below we have copied the review in full and provide our response in orange text.

Text quoted from the original manuscript is in grey and proposed changes based on the review are in blue.

We feel that thanks to these suggestions the manuscript will improve considerably and hope that our proposed revision will meet the criteria for publication in Climate of the Past.

Lukas Jonkers
On behalf of all authors.

The manuscript by Jonkers and colleagues compares multiple samples of the stable isotopes from the shells of the planktic foraminifer N. pachyderma from the same sediment trap samples. They then use a combination of nearby hydrographic records, modeling, and statistical analyses to assess the variability within a population not attributable to environmental factors, primarily temperature. They find a substantial amount of variability in multiple samples from the same cups, which is used to illustrate the inherent "excess" variability of reconstructions using very few shells. With increasing use of high resolution instrumentation making use of small samples and individual foraminifera analysis (IFA) more frequent, the implications of these findings are important.

I have a few suggestions which I hope the authors will find useful. My primary suggestion for the manuscript is to do with framing. From line 1 of the abstract, the rationale of the study is laid out to be an estimate of excess variability in individual shells measurements and therefore utility of IFA. The catch is that the methodology used here is not IFA but rather multiple pooled samples. Several assumptions are required to make the leap from environmental data and pooled measurements to an estimate of excess variability by a theoretical IFA measurement, some of which require additional justification. My comments include a few specific suggestions of where this may be helpful. However, it is also my opinion that the framing of this manuscript as a quantification of IFA excess variability may be slight overreach drawing from this particular dataset. There are certainly implications for IFA, and the rough calculation done here are useful in illustrating that. However, given the number of assumptions required and the use of pooled rather than individual shells in the analyses, overemphasis on a quantification of "noise" in IFA analyses specifically, may do a disservice to the really important findings of large excess variability.

We thank the reviewer for their constructive comments. We agree with the reviewer that our quantification of the excess variability requires more discussion and will add the following paragraph in section 3.3: "Whereas our modelling approach provides an estimate that is likely closer to reality than assuming that foraminifera reflect environmental conditions averaged over a single (calendar) month, our estimate could be evaluated by simulating other calcification trajectories. We found that our results are insensitive to the duration of chamber formation and experiments where we allowed complete shell formation within one day, equivalent to assigning all weight to the last chamber, yielded an expected 0.09 ‰ standard deviation of individual foraminifera δ18O. Therefore, the assumption of equal weight of the four chambers has little bearing on our results. Ultimately, the modelled foraminifera δ18O depends on the hydrographic data used to estimate δ18Oequilibrium. By using data from the surface and from great depth, we have obtained two end-member scenarios of vertical δ18Oequilibrium variability that implicitly encompass ontogenetic vertical migration. However, future estimates of expected individual foraminifera δ18O variability could be improved by explicitly incorporating horizontal δ18Oequilibrium variability and advection during shell growth in the modelling strategy.

Apart from being sensitive to our modelling design and data availability, our estimate of excess δ18O variability among individual shells is also sensitive to the quantification of variability among shells. To obtain a conservative estimate we excluded potential outliers. Were we to consider all measurements, the average standard deviation among groups would be 0.15 ± 0.11 ‰ (0.17 ± 0.09 ‰ during spring) and the resulting excess δ18O

variability 0.25 ± 0.19 ‰. Thus our approach yields a conservative and better constrained estimate of the excess variability."

We will also make sure to be more careful with our wording regarding the estimate of the excess noise in the abstract and in the conclusions. However, we think that our phrasing in the main text (e.g. "Assuming that our simulations are a reasonable approximation of reality, the excess variability (s.d.) that cannot be explained by variability in temperature and δ18Oseawater is therefore 0.11 ± 0.06 ‰.") is not overselling the results and we would prefer to keep the original text here.

Minor/specific points:

111: Why were outliers removed? Points that deviate farther from the mean would seem particularly valuable for this dataset, unless there is specific justification for their removal. Perhaps there is a reason for this data treatment that just needs to be better explained? This point was also raised by reviewer 2. The only reason to apply this filtering was to ensure that our analysis is insensitive to potential outliers, without making statements about the reliability of the removed data points. One could therefore view the variability in N. pachyderma stable isotope ratios that we use as a minimum, rendering our estimate of the magnitude of the excess variability conservative. We will make this reasoning clearer, both in the method section and in the discussion (see e.g. our suggested change above).

146: The assumption of chamber formation over one day in pachyderma is a bit misleading. While initial chamber formation may occur over one day (as in the referenced studies), calcification is likely more prolonged in this species. A better model than the spinose foraminifera observed in the Spindler and Be papers, might be congener N. dutertrei, where laboratory labelling experiments affirm that much of the calcite is added over a period of several days and nights as evidenced by banding and the apparently continuous uptakes of 'spikes' added in culture (see Fehrenbacher et al., 2017). We agree with the reviewer that our modelled chamber formation is an absolute minimum. It is, however, in agreement with the data from Spindler (1996) on N. pachyderma. We nevertheless checked what the effect is of longer chamber formation and reran our simulations with a four day duration of chamber formation as suggested for N. dutertrei (Fehrenbacher et al., 2017). The effect is negligible because of the high temporal autocorrelation in the d18Oequilibrium time series that renders the effect of smoothing insignificant. The expected standard deviation of foraminifera d18O based on our model is in both cases 0.08 permille. (Note that in our original submission we modelled chamber formation within at most one day and that yielded an expected standard deviation of 0.09 permille.) We will add the following text to section 2.3 to clarify this issue: "The assumed duration of chamber formation is based on culture studies (Bé et al., 1979; Spindler, 1996). However, culture studies in the closely related species N. dutertrei have shown that chamber formation may take up to four days (Fehrenbacher et al. 2017). Longer chamber formation could in theory reduce the variability foraminifera δ18O because of increased smoothing of the environmental signal. In practice this effect is however negligible because of strong temporal autocorrelation in the δ18Oequilibrium time series that renders the effect of smoothing of up to four days insignificant. Our approach thus yields an estimate of variability that is robust against the likely range of chamber formation duration."

281: I am struggling with this calculation, on which so much of the interpretation relevant to IFA rests. While this estimation accounts for the N term, it makes two assumptions. The first is that the sample mean would have been the same if IFA had been carried out rather that multiple pooled analyses – this is probably a reasonable assumption, if one has on minimal instrumental error and near identical calcite contribution from all shells. However, the other assumption is that the stable isotope value of an individual shell would be the same as the value of the pooled analyses. This is a less robust assumption, belied by even the conclusions of this paper. Individual shells would be expected to represent a greater range of values, and therefore overall greater deviation from the sample mean. I think the argument for calculating excess of theoretical IFA as such could benefit from a statement of these underlying assumptions.

The obvious rebuttal to the caveat(s) raised above is that these are necessary assumptions given the sample set and/or that once again the estimate of unexplained variance is highly conservative. This might be the case, but if so perhaps there is too much emphasis on the quantification of this speculative 0.19 per mill (and therefore 0.11 per mil) number as a noise threshold.

We appreciate the concerns by the reviewer and will better explain the way we performed the calculation. The reviewer is right about the first assumption that we assume an identical contribution to the total calcite mass for each shell (and hence identical mean values). We will state this more clearly. However, we do not make the second assumption. Instead, we explicitly derive the standard deviation among individual shells from the standard deviation of the pooled measurements, the former is - as the reviewer correctly notes - indeed larger (double in our case) than the latter. To clarify these issues we will change the sentence: "Since our measurements are based on groups of four shells, the standard deviation of individual shells is double (√4) the observed standard deviation." to: "Since our measurements are based on groups of four shells the observed standard deviation is an underestimate of the standard deviation among individual shells. Assuming that each shell in the group contributed equally to the total mass, the degree of underestimation of the standard deviation scales with the square root of the group size (Groeneveld et al. 2019). Thus we multiply the observed standard deviation by two (sqrt(4)) to obtain an estimate of the standard deviation of individual shells."

333-335: My reading of Livsey et al. (2020) is that lamellar and crust calcite were indistinguishable in d18O space
Good point, we accidentally mixed up Mg/Ca and d18O. This makes the likelihood that variable encrustation could explain the observed variability even smaller. We will delete the sentence and add: "However, the difference between crust and lamellar calcite δ18O of N. pachyderma intercepted in spring when the water column was well-mixed is not significant (Livsey et al. 2020). Variable encrustation can therefore not be the explanation for the excess δ18O variability observed during the isothermal conditions in spring."

Other minor points: I was curious about the lack of shell measurements here, as stable isotope values are well known to correlate with size, something that the authors discuss. I understand that this is a reanalysis and such measurements may no longer be available, but it is a point potentially worth addressing.
The reviewer rightly points out that size of individual shells would be an interesting parameter to have at our disposal. However, as the reviewer also correctly infers such

measurements are unfortunately not available. We would like to highlight though that we have analysed larger scale pattern in shell size and its influence on sedimentary stable isotope ratios in a previous paper (Jonkers et al., 2013).

References:

Fehrenbacher, J. S., Russell, A. D., Davis, C. V., Gagnon, A. C., Spero, H. J., Cliff, J. B., ... & Martin, P. (2017). Link between light-triggered Mg-banding and chamber formation in the planktic foraminifera Neogloboquadrina dutertrei. Nature communications, 8(1), 1-10.

Livsey, C. M., Kozdon, R., Bauch, D., Brummer, G. J. A., Jonkers, L., Orland, I., ... & Spero, H. J. (2020). Highâ resolution Mg/Ca and δ18O patterns in modern Neogloboquadrina pachyderma from the Fram Strait and Irminger Sea. Paleoceanography and Paleoclimatology, 35(9), e2020PA003969.

References

Fehrenbacher, J. S., Russell, A. D., Davis, C. V., Gagnon, A. C., Spero, H. J., Cliff, J. B., Zhu, Z., and Martin, P.: Link between light-triggered Mg-banding and chamber formation in the planktic foraminifera *Neogloboquadrina dutertrei*, Nature communications, 8, 15441, 2017.

Jonkers, L., van Heuven, S., Zahn, R., and Peeters, F. J. C.: Seasonal patterns of shell flux, $\delta^{18}O$ and $\delta^{13}C$ of small and large *N. pachyderma* (s) and *G. bulloides* in the subpolar North Atlantic, Paleoceanography, 28, 164–174, 2013.

Spindler, M.: On the salinity tolerance of the planktonic foraminifer *Neogloboquadrina pachyderma* from Antarctic sea ice, Proceedings of the NIPR Symposium on Polar Biology, 9, 85–91, 1996.

---

## Author Comment (AC3)

We would like to thank the reviewer for their careful reading of our manuscript and their constructive comments. Below we have copied the review in full and provide our response in orange text.

Text quoted from the original manuscript is in grey and proposed changes based on the review are in blue.

We feel that thanks to these suggestions the manuscript will improve considerably and hope that our proposed revision will meet the criteria for publication in Climate of the Past.

Lukas Jonkers
On behalf of all authors.

The manuscript submitted by Jonkers et al. describes a study (based on existing data from earlier publications) that aims to assess whether planktic foraminifera of the genus Neogloboquadrina pachyderma accurately record environmental parameters (here: temperatures deduced from d18O, and d13C). Shells of N. pachyderma were derived from a sediment trap, moored in the Irminger Sea. The trap collected sinking plankton during multiple years, and the collection intervals were roughly 2.5 weeks. For analysis, Jonkers et al. pooled four N. pachyderma shells from each sample vial, and multiple groups of four shells were analyzed for each collection interval. A within-sample variability of 0.11‰ for d18O and of 0.10‰ for d13C was found, independent of the season or month of sampling. Furthermore, the variability in d18O and d13C exceeds water column variability in spring when the water column is isothermal.

In order to assess potential sources for this variability, the authors run simulations (main parameters are the potential timespan of chamber formation, calcification depth, and delay due to settling), and conclude that the observed variability in d18O can only partially be explained by environmental variability. The authors estimate an "excess noise" on d18O of about 0.11‰ (biological or other yet unknown origin), which, as the authors postulate, needs to be taken into account when interpreting geochemical variability among individual foraminifera.

This is an interesting study/manuscript that is certainly an important contribution, however, there are certain issues that the authors should address:

(1) Jonkers et al. is linking this study to Individual Foraminifera Analysis (IFA), which is increasingly common with the rapid development of new or improved analytical approaches. However, IFA are, senso stricto, measurements of single, individual foraminifera shells. However, the authors were analyzing groups of four shells. I am not sure to what extent the findings of Jonkers et al. can be interpolated to 'true' single-shell IFA, but I would prefer to remove all references to IFA or soften the wording. However, Jonkers et al. raise an important question: We need to decide between the "reliability" of individual planktic foraminifera shells as a proxy recorder, and the potential attenuation of high-frequency or short-lived climate signals due to the measurement of populations that are too large to record these short-term signals. Instead of referring to IFA, I recommend to include a short discussion about sample sizes for paleoclimate records (built upon Schiffelbein and Hills, 1984, and subsequent studies). There is no simple answer – but the new data presented by Jonkers et al. provide the opportunity to discuss this topic from a new/different perspective. The reviewer raises some important points. We understand the doubts by the reviewer, but do not agree that our analyses have no implications for the interpretation of IFA results. The replicate measurements on groups of four shells of course cannot directly provide information about the stable isotope variability among individual shells. However, what these replicate measurements can provide is an estimate of variability within the population of planktonic foraminifera that cannot be obtained from a single measurement (whether on a sample containing many foraminifera, or single shells). This estimate of the variability within the population, even though an underestimation of the variability among individuals, is valuable knowledge for the interpretation of IFA results. We are therefore convinced that the framing of our study along the lines of "implications for IFA" is justified. We make our reasoning clearer in a revised version and will add the following sentence to section 2.2: "Even though the measurements were done on groups of four shells, the replicate

measurements on small numbers of shells allow us to obtain a first order estimate of the minimum stable isotope variability within the population of N. pachyderma. Our analyses are therefore meaningful for the interpretation of IFA results."

We would also like to stress that all our modelling exercises are consistent with the measurements on groups of shells, not on individual shells. It is only in section 3.3 that we provide an estimate of the inter-individual variability. This estimate is based on the mathematical relationship between the standard deviation among groups (of shells) and the standard deviation among individuals (shells) that make up those groups. Assuming that each shell contributes equally to the total mass of the group, the standard deviation in the d18O of individual shells scales with the square root of the group size, in our case sqrt(4) = 2. We realise that this calculation was not described clearly enough and will change the wording in section 3.3. Specifically, we will change: "Since our measurements are based on groups of four shells, the standard deviation of individual shells is double (√4) the observed standard deviation." to: "Since our measurements are based on groups of four shells the observed standard deviation is an underestimate of the standard deviation among individual shells. Assuming that each shell in the group contributed equally to the total mass, the degree of underestimation of the standard deviation scales with the square root of the group size. Thus we multiply the observed standard deviation by two (sqrt(4)) to obtain an estimate of the standard deviation of individual shells."

The argument above also explains why our estimate of IFA is more robust than what could be obtained from analyses that pooled more specimens. For example, variability obtained from replicates of pooled analyses of 25 shells scales to the IFA variability by a factor of 5. This means that to constrain the IFA variability as well as in analyses of 4 shells, one would need 2.5 x as many replicates.

The reviewer suggests that instead of focussing on the implications for IFA, we should consider discussing the number of foraminifera that should be analysed for robust palaeoceanographic reconstructions. We agree that this is an important topic. However, the variability among sedimentary foraminifera stable isotope ratios depends on many more factors than we can assess from our time series. For instance, it depends on the seasonal amplitude of temperature (and d18Osw) variation, the seasonality in the flux of foraminifera, the sedimentation rate as well as the bioturbation depth. The effect of these factors has been discussed extensively in the literature (e.g. Dolman and Laepple, 2018; Jonkers and Kučera, 2017; Lougheed and Metcalfe, 2021). In our opinion such a discussion would go beyond our original question about the reliability of single planktonic foraminifera shells as recorders of environmental conditions.

(2) On purpose, the authors excluded the possibility of horizontal drifting -which is okay. Including horizontal drifting will add new layers of complexity and uncertainties, and potentially raise a whole new set of open questions and challenges. Still, horizontal drifting should be discussed as a potential source of the large measured d18O variability in N. pachyderma shells that exceeds the annual range in "d18O equilibrium" values at the location of the sediment trap. In quickly checking the velocities within the Irminger Gyre (e.g., Våge et al., 2011), the shells can be transported to the sediment trap over significant distances and "import" proxy-signals from a very different location. Basically, the authors exclude horizontal drifting, run the model, and postulate that the measured d18O (and d13C) data in the shells cannot be reproduced with local temperature and d18Osewater data,

independent of the selected calcification depth. Thus, the authors ascribe the 'excess' variability in foraminifera d18O and d13C to biological (and/or other) factors. Latest at this point, horizontal drifting should be again included into the discussion (although it was not included in the model, which is okay).

The reviewer raises a fair point that indeed deserves more discussion. The possibility of advection is real, even though the influence on the stable isotope variability is not directly scalable with the advection distance. We tried to allude to this in the last paragraph of section 2.3, but we will also include a more extensive discussion about our estimate of excess variability. We propose to add: "Whereas our modelling approach provides an estimate that is likely closer to reality than assuming that foraminifera reflect environmental conditions averaged over a single (calendar) month, our estimate could be evaluated by simulating other calcification trajectories. We found that our results are insensitive to the duration of chamber formation and experiments where we allowed complete shell formation within one day, equivalent to assigning all weight to the last chamber, yielded an expected 0.09 ‰ standard deviation of individual foraminifera δ18O. Therefore, the assumption of equal weight of the four chambers has little bearing on our results. Ultimately, the modelled foraminifera δ18O depends on the hydrographic data used to estimate δ18Oequilibrium. By using data from the surface and from great depth, we have obtained two end-member scenarios of vertical δ18Oequilibrium variability that implicitly encompass ontogenetic vertical migration. However, future estimates of expected individual foraminifera δ18O variability could be improved by explicitly incorporating horizontal δ18Oequilibrium variability and advection during shell growth in the modelling strategy.

Apart from being sensitive to our modelling design and data availability, our estimate of excess δ18O variability among individual shells is also sensitive to the quantification of variability among shells. To obtain a conservative estimate we excluded potential outliers. Were we to consider all measurements, the average standard deviation among groups would be 0.15 ± 0.11 ‰ (0.17 ± 0.09 ‰ during spring) and the resulting excess δ18O variability 0.25 ± 0.19 ‰. Thus our approach yields a conservative and better constrained estimate of the excess variability." to section 3.3.

(3) A puzzling observation is the fact that some group of four shells feature significantly higher d18O values than we would expect at sample location, even when we assume calcification during the coldest season and at a large water depth (see Fig. 2). This is an interesting finding and should be discussed. Low d18O values in N. pachyderma are often observed, and some previous studies (e.g., Bauch 1997, Ravelo and Hillaire-Marcel (2007), Simstich et al., (2003)…) postulated that either vital effects or the effect of low-d18O meltwater lenses cause low d18O values in N. pachyderma shells. However, reports of N. pachyderma shells that are "too heavy" in their d18O composition are rare. Were the shells transported from colder waters to the location of the sediment trap? This should be further discussed. In particular, it needs to be emphasized that each data point integrated the composition of four shells. Thus, the spread of individual shells in d18O (and d13C) is likely larger, and single shells may feature even higher d18O values than the group of four. If it is not possible to reconstruct these high d18O from the water column profile – what is the explanation, if we exclude horizontal drifting?

The reviewer touches on an interesting point. Apart from d18O values lower than equilibrium, which we attributed to remnants of the summer population that survived without

calcifying (Jonkers et al., 2010) there are indeed also samples with a d18O higher than d18Oeq. We agree that these data points are puzzling.
The reviewer suggests that advection from colder waters could be an explanation. We agree that this could be the case. However, advection from colder water likely means advection from the East Greenland Current, which is also considerably fresher and hence has lower d18Oseawater (around -2.5 permille VSMOW). The d18Oseawater effect would therefore overwhelm the temperature effect and advection from the East Greenland Current is therefore likely to lead to lower foraminifera d18O.
We note that during the time when d18Ocalcite higher than d18equilibrium is observed, some samples also show lower than equilibrium values (Fig. 2B). This large spread in the d18Ocalcite is entirely consistent with the hypothesis that foraminifera d18O contains additional, temperature and d18Oseawater-independent, noise and are therefore individually not reliable as environmental indicators.

(4) For this study, defining criteria for outliers is very important and critical. The authors defined outliers as being more than 1.5 times the interquartile range away from the overall mean. Was this selection arbitrary? Do we know whether the "outliers" provide a true signal? Four shells are measured together, thus, one or two shells within this group of four must feature very different d13C or d18O values to shift the averaged composition of four shells sufficient to trigger the 'outlier' criterion. Jonkers et al. removed 6% of the d18O data. This is a high number. In other words: It seems the authors believe that 6% of all d18O measurements conducted within the framework of this study are not trustworthy. This needs to be discussed in more detail. The sample material was clean and well preserved (sediment trap, no issues with clay contamination or diagenesis), and standard procedures/equipment was used for sample preparation and analysis. We have many decades of experience with this analytical approach. Thus, in theory, the quality of the data should be as good as it can get. But 6% removed???
We agree with the reviewer that this is a point that requires further explanation. Our rationale to apply a strict outlier criterion was to avoid discussion about the influence of potential outlier effects on our interpretation. We did not want to imply that measurements identified as outliers are unreliable. Please also note that this approach is fairly common in IFA studies (Ganssen et al., 2011; Groeneveld et al., 2019). We will add the following sentence to section 2.2 where the outlier removal is described: "In order to obtain a conservative estimate of the variability among the measured groups of N.pachyderma shells we remove possible outliers."

Importantly, even when reducing the variability by removing samples outside 1.5 times the interquartile range, the remaining variability in stable isotope ratios is still larger than what could be expected. Our conclusions on the excess variability are therefore conservative. We will elaborate more on this in the discussion on the quantification of the excess variability (see the proposed text under point 2 above).

(5) General comment regarding the figures: Many labels in the figures are too small. It is okay when reading the publication as PDF (which most of us will do), but much information will be lost when the figures are printed. In addition, the manuscript would greatly benefit from some careful 'wordsmithing'.

We will increase the font size in the figure labels and do our best to improve the writing. We would like to thank the reviewer again for their careful reading and the many helpful suggestions to improve the text.

Some minor suggestions:

Line 54: The sentence seems to be incomplete. Suggestion for completion: "…and only few consider calibration issues associated with individual planktic foraminifera (Glaubke et al., 2021) as a source of uncertainty".
Will do.

Line 56, 57: "geochemistry is too generic". Temperature exerts a first order control on Mg/Ca and d18O (when d18Osw is accounted for). There are several other foraminifera-based proxies that are not primarily controlled by temperature.
Good point; we will replace "geochemistry" with "Mg/Ca and d18O".

Line 67: consider rewording: a proxy is only approximating a parameter of interest. It is not a "precise" environmental indicator. Precise implies precision. Better choices are 'robust', or 'reliable'.
We will change to "reliable".

Lines 78-83- the last paragraph of the introduction describes results or conclusions (…"We observe marked variability… … and find that the observed variability….. … we argue that this biological…"). I leave it up to the authors, however, I strongly recommend keeping the introduction descriptive, without mentioning the results or even some interpretation
We would prefer to keep this as it is as we think it makes for more interesting reading if the editor agrees.

Line 101: Can the authors provide more detail? 45 samples (= collection intervals) were analyzed, most of them were measured at least twice. However, it follows from Section 2.1 that the sediment trap provided much more than 45 samples (or collection intervals). It would be nice if the authors could provide more information about the criteria for sample selection.
To make it clearer that not all samples (collection intervals) contained foraminifera we will add the following sentence: "Not all samples from the time series contained enough shells of N. pachyderma (Fig. 1), so the complete data set consists of 172 measurements from 45 samples, of which 163 are from 36 samples with at least two measurements."

Line 106: I am a bit confused. I thought IFA stands for "Individual Foraminifera Analysis", which means individual shells. However, according to Section 2.2, groups of four N. pachyderma shells were analyzed. Thus, the number of shells is high compared to IFA, not low, as stated by the authors. I am not even sure if groups of four shells can or should be considered as IFA.
We are sorry that the reviewer got the impression that we performed individual foraminifera analyses. We never intended to claim that we did, and will make it clearer that our measurements were done on small groups of shells. We will add the following sentence "Even though the measurements were done on groups of four shells, the replicate measurements allow us to obtain a first order estimate of the minimum stable isotope variability within the population of N. pachyderma. Our analyses are therefore meaningful for

the interpretation of IFA results." to section 2.2. We understand the confusion about the number of shells per sample, the crucial difference is in the number of replicates: for IFA usually in the order of 50-70, whereas we have used up to six replicates per sample. To avoid confusion we will replace "The number of shells measured per sample …" with "The number of replicate measurements per sample…"

This comment echoes the first comment by the reviewer and we would like to emphasise that we do not consider our measurements equivalent to single shell analysis, but that the conclusions we derive from our data are still important for the interpretation of IFA data, precisely because the observed variability among groups of four shells represents a minimum estimate of the variability among individual shells. In section 3.3 we will also elaborate further on how we derive an estimate of the variability among individual shells from the measurements on groups of shells.

Line 107: "weeks to month" – does this refer to the collection intervals, or the combination of collection interval + life span of the foraminifera (in particular the time when they grew their shells)? I think this should be mentioned for clarity.
To make this issue clearer we will change "This is however justified given the short integration time of sediment trap samples (in our case 16-19 days) compared to sediment samples (at least decades to centuries)." to "This is however justified given the short collection intervals of sediment trap samples (weeks to months) compared to the long integration time of sediment samples (at least decades to centuries)."

Line 120: Yes, but there are also studies postulating that N. pachyderma features a (negative) vital effect in d18O (Bauch, Simstich, Hillaire-Marcel, and many others). Although I am okay how this is written, adding a short discussion – emphasizing why the authors believe that N. pachyderma calcifies without a vital effect for d18O – would be helpful
The reviewer touches on an interesting point. The offset, or lack thereof, from d18Oequilibrium is important to constrain when interpreting the d18O of foraminifera. However, for the present study, the issue is only of limited relevance. This is because in the temperature range investigated here, the slopes of the different palaeotemperature equations are nearly identical (Jonkers et al., 2013) and the effect of using a different equation on the d18O variability is thus negligible. For this reason we deem a discussion about a possible offset from d18Oequilibrium more a distraction than an addition in this manuscript. The reader is referred to two publications where this issue is elaborated. A potential reason for the difference with other studies (including those mentioned by the reviewer) is the use of different palaeotemperature equations.

Line 121: It shall read "Jonkers et al., 2010, 2013". Same in line 121
OK.

Lines 122-125: please reword the sentence – overuse of 'because' (we use these because…and because )
Will do.

Line 127: I think it shall read "regressions" (plural)
We disagree, multiple (linear) regression is regression with multiple predictor variables, not multiple regressions with single predictor variables.

Line 130: what does "available as climatology" mean? Same line: Use "spatial resolution" instead of "same level of detail"?

Climatology data means the long-term average as for instance provided in the world ocean atlas. The lack of detail is hence not only in space, but more importantly in time (see fig. 2). To clarify, we will replace: "δ13CDIC data are available as climatology only and can hence not be used to the same level of detail as δ18O" with "Since the δ13CDIC data are derived from data that represent long-term average conditions (climatology), they cannot be used to the same level of detail as δ18O"

Lines 130, 131: measured variability in foraminiferal d13C (to make it clearer)?
Will do.

Line 135: It sounds as if the formation of the entire shell takes place in the same water depth. Most planktic foraminifera (also N. pachyderma) migrate to deeper waters as part of their ontogenetic development. Earlier chambers are typically formed in shallower waters than the later chambers (and crust, if present). This should be mentioned here.

The reviewer is right about the possibility of ontogenetic vertical migration in N. pachyderma. However, we want to discuss - and model - the temporal and spatial integration aspects separately. We want to first highlight the temporal aspects of calcification (this sentence) and we discuss variability in calcification depth, including vertical migration in lines 170 to 180.

Line 142: What does "survival' in the water column (without calcification)" mean? The last chamber is formed, the organism is not further calcifying (end of life cycle), and the finished shell is sinking without further modification (calcification or dissolution) to the trap. Why 'survival'?

The reviewer assumes that the foraminifera die immediately after formation of the last chamber, this need not to be the case in culture studies (Spindler, 1996) and we explicitly take this possibility into account. We also model this behaviour because of indications for the presence of a non-calcifying population in the data themselves. In our first paper describing these data (Jonkers et al., 2010) we attributed the low d18O values in autumn to remnants of the summer population that were no longer actively calcifying. However, we do see that the term "surviving" may be confusing and will change "add a delay between formation of the final chamber and arrival at the sediment trap that reflects survival in the water column (without calcification) and sinking time." to "add a delay between formation of the final chamber and arrival at the sediment trap that reflects time spent in the water column without calcification as well as sinking time." We will also change the wording elsewhere in the manuscript (section 3.2).

Line 158: For clarification: The authors mean the time span between the formation of the first chamber of the final whorl, and the last chamber of the final whorl? – please reword for clarity

We will make it clearer that this refers to "modelled" foraminifera that, as stated, consist of only four chambers. In this sense, this has no bearing on real foraminifera shells that consist of multiple whorls. We will change the text to: "calcification spans (the time it takes to form the four-chambered synthetic shell chambers)"

Line 181: "ignore" sounds very harsh. What about: "…was not considered…"

Will change.

Line 186: For clarity: What about: "In order to approximate the measured d18O values with our model simulation, we average the d18O of four simulated shells"…
Will change.

Line 191: For clarity: … if the standard deviation of the measured d18O values (correct?) is higher than the observed…
Will change.

Lines 195, 205, 209: please do not use "ignore"
We will reword.

Line 203: please reword "foraminifera would see". What about: "the additional variability in temperature the individual planktic foraminifera would be exposed during its life cycle"
Will change.

Line 224: suggestion: "…and the range in measured d18O is, in all cases, smaller than the….". However, this is a bit confusing. If I understand correctly, the range in measured d18O is consistently smaller than the seasonal range in surface d18O equilibrium. However, the range in measured d18O exceeds the range of d18O equilibrium during time intervals with an isothermal water column (see lines 243-245). The authors may consider to put these information together for clarity..
We will change "There is no relationship between the number of measurements within a sample and the range in δ18O and it is always smaller than the seasonal range in surface δ18Oequilibrium and most of the time also smaller than the vertical gradient in δ18Oequilibrium (Fig. 4)." to "There is no relationship between the number of measurements within a sample and the range in δ18O (Fig. 4). The within sample range is always smaller than the seasonal range in surface δ18Oequilibrium. Most of the time the observed δ18O range is also smaller than the vertical gradient in δ18Oequilibrium, except during isothermal conditions in spring when it exceeds the δ18Oequilibrium range  (Fig. 4)."

Line 230: suggestion: "…regarding these initial observations…"
OK

Line 235: suggestion: "The fact that this cannot be seen in the data…"
OK

Line 239: "if the observed variability in foraminifera d18O is higher…. expected from temperature and d18O seawater at the time…"
Will reword to

Line 240: prior to the sampling
We are unsure what the reviewer is commenting about. These words appear exactly like this in the text.

Line 241: delays (plural). Please see my earlier comment regarding 'survival'. I still don't know what it actually means. I assume the authors would like to say that the 'finished' shells

remains in the water column without any further modification (of course, these are assumptions for the model, nature is more complex), until it is collected in the sediment trap
We will add the "s". Please see our response above regarding survival.

Line 246: for clarity: please mention again: what are the two scenarios? (1) Variable calcification depth, and (2) calcification during summer?
Will change "Our simulations are thus sensitive to the choice of calcification depth and it is important to assess if both scenarios are equally realistic. We can do so by determining the prediction error in the mean δ18O across all samples (Fig. 6)." to: "Our simulations are thus sensitive to the choice of calcification depth and it is important to assess if the scenario with variable depth habitat is more realistic than the scenario with constant, near-surface habitat. We can compare both scenarios by determining the prediction error in the mean δ18O across all samples (Fig. 6)."

Line 273: It shall read "Davis et al., 2017, 2020a"
OK

Line 286: suggestion: "when variations in temperature and…"
We will change the wording.

Line 296: "In the first study, the range in …amounts to 0.15‰ (Leduc et al., 2009). In the second study,…"
OK

Line 321-326: Please also add a few sentences explaining that N. pachyderma features no symbionts, thus, we can exclude the effect of symbiont activity on shell-d13C
The reviewer rightly points out that symbiont activity cannot affect the d13C in N. pachyderma. However, in this section we discuss possible causes for variability in both d18O and d13C. Since symbiont activity does not affect d18O we see no merit in mentioning factors that could only affect d13C.

Line 332-325: This is an important discussion – the proportion of crust to lamellar calcite. The authors are discussing that the crust calcite has a different d18O value than the lamellar calcite (lines 339-340). Yes, but this is because the crust is typically formed in deeper waters. Livsey et al. (2020) has shown that both the crust and the lamellar calcite likely form in equilibrium with ambient temperature and seawater d18O.

Therefore, the difference between lamellar calcite d18O and crust calcite d18O can only be explained by downward migration in the water column. However, in this manuscript, the authors postulate that the calcification depth of N. pachyderma is limited to a well-defined, narrow band. There is the risk that this discussion is contradicting previous statements from the authors.
We agree with the reviewer that this is an important point that should be clarified. Reviewer 1 also pointed to an inconsistency that we inadvertently included. We apologise for the confusion.

Livsey et al measured the d18O on lamellar and crust calcite from N. pachyderma shells from the same sediment trap time series. They performed measurements on shells

intercepted in spring, i.e. formed during isothermal conditions and found a small, but not significant, difference between crust and lamellar calcite d18O. The conclusion put forward by the reviewer ("Therefore, the difference between lamellar calcite d18O and crust calcite d18O can only be explained by downward migration in the water column") is not supported by the data and therefore not in conflict with the inferred narrow band of calcification. We propose to change the section to: "The excess variability could also arise from differences in the proportion of crust to lamellar calcite. Variable crust to lamellar calcite ratios among foraminifera could therefore add temperature-independent noise, similar to what has been suggested for Mg/Ca (Jonkers et al., 2021, 2016). However, the difference between crust and lamellar calcite δ18O of N. pachyderma intercepted in spring when the water column was well-mixed is not significant (Livsey et al. 2020). Variable encrustation can therefore not be the explanation for the excess δ18O variability observed during isothermal conditions in spring. In addition, this explanation would require that the crust and lamellar calcite also have different carbon isotope ratios. However, previous work is inconclusive in this regard. Observations from plankton hauls suggest that encrusted and crust-free N. pachyderma have systematically different δ13C, but that the effect of encrustation is not as strong as on δ18O (Kohfeld et al., 1996). A larger dataset from the sediment on the other hand, indicates no effect of encrustation (Healy-Williams, 1992). Whether or not variable encrustation is the cause of the observed excess variability in δ18O and δ13C therefore remains an open question." to avoid confusion.

In addition, there is no discussion whether the authors have carefully investigated the shells by binocular microscope. N. pachyderma shells collected by sediment traps typically feature only a thin crust, or the crust is entirely absent (in contrast, fossil shells typically feature a thick crust). I believe that some information about the degree of encrustation of the investigated N. pachyderma shells would help to bolster the discussion regarding the potential impact of crust calcite on the variability in d18O.
We have not systematically investigated the degree of encrustation in the time series, however, no crust-free specimens were observed (see also Jonkers (2016)).

Line 350: I prefer to be careful and not implying that this is the case for all planktic foraminifera. So far, we only have data for N. pachyderma. For other species, there are only indirect indications.
We disagree and think our findings have broader implications than for N. pachyderma alone, especially because of the indications for similar variability in other species (see lines 366-368 in the original text). We have also stressed the need for more research (lines 369-360) and feel that our phrasing ("we therefore presume…") is sufficiently careful. Thus we prefer to keep to the original wording.

Line 356: Thus, for now, it needs to be assumed that N. pachyderma forms it shell in equilibrium with seawater d18O and ambient temperature, superimposed by a noise of 0.11‰? I still would be a bit more cautious. The model simplifies very complex natural processes, and some of the apparent excess noise may reflect inabilities of the model to accurately reflect nature. Culture studies would help to provide more confidence (of course, there is the issue of culturing N. pachyderma successfully...)
We agree that the quantification of the noise level model-dependent and will make this clearer in the text (see also the comment by reviewer 1). We also agree that culture studies may help and would like to highlight that there has been tremendous progress in culturing

this species recently (Davis et al., 2020). We are therefore happy to mention this again in the text.

Line 373: Please add more information regarding Mg/Ca (temperature proxy, why could it be useful in future studies to elucidate the cause of variability). Without additional information, this may not be clear to some readers.
We will add this information.

Line 385: "…that has so far been…"
Ignored? We will change to " … has so far not been considered…"

Fig. 2: Although mentioned in the figure caption, it would be nice to have a legend, explaining yellow points and green bars. Please add a description of Panels B) and D) to the figure caption. These enlarged plots show the sampling interval April 2006 – March 2007, correct?
We will add the following sentence to the figure caption to better explain what is shown in the different panels: "Panels A and C show the time series of $\delta 18O$ and $\delta 13C$, respectively. Panels B and D highlight the annual pattern, they show the same data collapsed onto a single year.". We will also move the sentence explaining the meaning of the colours, so the explanation appears earlier in the caption.

Fig. 4: I cannot see any difference between the lines in grey color, depicting the difference in d18O between the surface and 200-250 m water depth, and the (same?) line in Fig. 2 depicting surface d18Oeqilibrium.
The difference between surface and deep d18Oeq shown in figure 4A is indeed very similar, but not identical, to the line showing surface d18Oeq in Fig. 2. This is because of the lack of substantial variability of d18Oeq at depth (dark line in Fig 2).

References:

Bauch, D. (1997). "Oxygen isotope composition of living Neogloboquadrina pachyderma (sin.) in the Arctic Ocean." Earth and Planetary Science Letters 146: 47-58.

Ravelo, A. C. and C. Hillaire-Marcel (2007). The Use of Oxygen and Carbon Isotopes of Foraminifera in Paleoceanography. Developments in Marine Geology. H. M. Claude and V. Anne De, Elsevier. Volume 1: 735-764.

Simstich, J., et al. (2003). "Paired d18O signals of Neogloboquadrina pachyderma (s) and Turborotalita quinqueloba show thermal stratification structure in Nordic Seas." Marine Micropaleontology 48: 107-125.

Våge, K., Pickart, R.S., Sarafanov, A., Knutsen, Ø., Mercier, H., Lherminier, P., van Aken, H.M., Meincke, J., Quadfasel, D., Bacon, S., 2011. The Irminger Gyre: Circulation, convection, and interannual variability. Deep Sea Research Part I: Oceanographic Research Papers, 58(5): 590-614.

References

Davis, C. V., Livsey, C. M., Palmer, H. M., Hull, P. M., Thomas, E., Hill, T. M., and Benitez-Nelson, C. R.: Extensive morphological variability in asexually produced planktic foraminifera, Science Advances, 6, eabb8930, 2020.

Dolman, A. M. and Laepple, T.: Sedproxy: a forward model for sediment-archived climate proxies, Climate of the Past, 14, 1851–1868, 2018.

Ganssen, G. M., Peeters, F. J. C., Metcalfe, B., Anand, P., Jung, S. J. A., Kroon, D., and Brummer, G. J. A.: Quantifying sea surface temperature ranges of the Arabian Sea for the past 20 000 years, Climate of the Past, 7, 1337–1349, 2011.

Groeneveld, J., Ho, S. L., Mackensen, A., Mohtadi, M., and Laepple, T.: Deciphering the Variability in Mg/Ca and Stable Oxygen Isotopes of Individual Foraminifera, Paleoceanography and paleoclimatology, 34, 755–773, 2019.

Jonkers, L. and Kučera, M.: Quantifying the effect of seasonal and vertical habitat tracking on planktonic foraminifera proxies, Climate of the Past, 13, 573–586, 2017.

Jonkers, L., Brummer, G.-J. A., Peeters, F. J. C., van Aken, H. M., and De Jong, M. F.: Seasonal stratification, shell flux, and oxygen isotope dynamics of left-coiling *N. pachyderma* and *T. quinqueloba* in the western subpolar North Atlantic, Paleoceanography, 25, PA2204; 10.1029/2009PA001849, 2010.

Jonkers, L., Jiménez-Amat, P., Mortyn, P. G., and Brummer, G.-J. A.: Seasonal Mg/Ca variability of *N. pachyderma* (s) and *G. bulloides*: Implications for seawater temperature reconstruction, Earth and planetary science letters, 376, 137–144, 2013.

Jonkers, L., Buse, B., Brummer, G.-J. A., and Hall, I. R.: Chamber formation leads to Mg/Ca banding in the planktonic foraminifer *Neogloboquadrina pachyderma*, Earth and planetary science letters, 451, 177–184, 2016.

Lougheed, B. and Metcalfe, B.: Testing the effect of bioturbation and species abundance upon discrete-depth individual foraminifera analysis, , https://doi.org/10.5194/bg-2021-202, 2021.

Spindler, M.: On the salnity tolerance of the planktonic foraminifer *Neogloboquadrina pachyderma* from Antarctic sea ice, Proceedings of the NIPR Symposium on Polar Biology, 9, 85–91, 1996.